# Formulation of Development Strategies for Regional Agricultural Resource Potential: The Ukrainian Case

Nestor Shpak [1], Ihor Kulyniak [1], Maryana Gvozd [1,*], Jolita Vveinhardt [2] and Natalia Horbal [1]

[1] Institute of Economics and Management, Lviv Polytechnic National University, 79013 Lviv, Ukraine; nestor.o.shpak@lpnu.ua (N.S.); ihor.y.kulyniak@lpnu.ua (I.K.); natalia.i.horbal@lpnu.ua (N.H.)
[2] Faculty of Applied Sciences, WSB University, 41-300 Dąbrowa Górnicza, Poland; jvveinhardt@wsb.edu.pl
* Correspondence: mariana.y.hvozd@lpnu.ua

**Abstract:** The agricultural sector is one of the leading ones in the economy of many countries, as it creates the basis for their economic growth. Every region in every country has its own unique sphere of social reproduction due to different resource potential, i.e., fertile soils, favourable climatic conditions, etc. Under such conditions, it is irrelevant to choose a single development path for this sector. Given these facts, the study formulates development strategies for regional agricultural development of the country. In particular, we grouped regions into clusters according to the level of development of crop production and stockbreeding potential, as well as investment attractiveness. The method of cluster analysis was used to group regions by the level of capacity development, whilst the matrix method was used to formulate a matrix for choice of the strategy to improve the agricultural potential of regions. On the basis of the conducted analysis, the 3D matrix for the choice of investment strategy of the regional agricultural development of the country depending on the level of development of crop production and stockbreeding potential and investment attractiveness was constructed.

**Keywords:** agriculture; region's agricultural resource potential; strategy; cluster analysis; matrix method; crop production

## 1. Introduction

Currently the agricultural sector is an important indicator of economic development in many countries. The development of rural areas largely depends on the activities of local agricultural enterprises [1]. It relates to both developed and developing countries. Rural development plays a crucial role in EU cohesion policy [2]. Given this fact, the European Union launched the National Rural Development Program (RDP); one of its aims is the establishment of agricultural producer organisations to assist cooperation among small- and medium-sized farms and thus to improve their performance [3]. However, the development of rural areas largely depends on the activities of local agricultural enterprises [4]. This is especially important for Ukraine, as its agricultural sector occupies a prominent place among other industries [5]. In this context, considerable attention of scholars and practitioners is paid to the development of the resource potential of the agro-industry. In addition, one should remember that the development of agricultural enterprises touches upon sensitive social, economic and ecological issues [6]. The researchers even note that large agricultural holdings in Ukraine take the form of "economics in the economy" [7] which is associated with some—in many cases negative—consequences, such as the limited impact of the state on those entities. These facts confirm the complexity of the phenomenon in reality in the context of Ukraine.

The increase of the volume of agricultural production remains the main strategy for food supply in Ukraine [8]. As practice shows, in Ukraine the main problem is the development of agricultural enterprises. It is the methods and results of solving this

problem that determine the efficiency of agricultural enterprises, the competitiveness of their products [9] and their further development.

The issue of the structural balance of resource potential, in particular the relationship between its individual components in their cost form [10], needs to be addressed. It relates, inter alia, to the structural proportions between crop and livestock land fund and the availability of equipment for tillage, stockbreeding and feed production, etc. The imbalance in these proportions leads to a decrease in the efficiency of agricultural enterprises.

Priority sectors of agriculture are crop production and animal husbandry (stockbreeding). The main purpose of crop production is the processing of cultivated plants for the production of crop products as well as providing the population with food, stockbreeding with feed, and industry with raw materials of plant origin. The purpose of animal husbandry is to breed farm animals for production. Its products are food for humans (meat, milk and dairy products, eggs, etc.) and raw materials for food and light industry. The efficient and successful functioning of the livestock and crop industries needs in-depth analysis in order to improve the country's national economy, quickly integrate it into the world economic system and improve living standards and well-being, especially in rural areas, by creating small farms [11]. Ukraine, in particular, has a high level of resource potential based on fertile soils and favourable climatic conditions. However, there are still opportunities to improve the quality, productivity, profitability, investment and innovation attractiveness of the sector. It is important to emphasise that each region of Ukraine has its own unique sphere of social reproduction, which encourages the diversification of the ways of the development of this sector for the country as a whole. Agriculture is the main driving force for the development of the country's economy and ensuring the welfare of the population. Unfortunately, for the last few years the development of the agricultural sector has been suspended due to insufficient and ineffective state support. Under such conditions, the priority is to create recommendations for public intervention aimed at promoting models of agro-industrial development [12]. One of the problems that needs to be addressed immediately is the inefficient investment support of agricultural sectors. A well-established system of primarily capital investment can help in the stabilisation of the industry and the progressive development of its potential.

Intensifying Ukraine's involvement in global world economic processes requires increasing attention to agricultural development prospects in order to expand opportunities to ensure its food security. Rationality and maximum efficiency of the use of resources involved in the process of agricultural production is a necessary prerequisite for achieving a balanced interest of society in the social, economic and environmental spheres. It is also the foundation for ensuring national priorities in the development of the agricultural sector, in particular the attainment of food security, creating successful export policies, reaching profitability goals for agricultural producers [3] and ensuring a socially-oriented state agricultural policy [13]. The priority areas of increasing the level of investment attractiveness of the agricultural sector of Ukraine are the selection of an innovative development strategy and attracting investment in high-tech areas of agricultural production as well as improving the organisational and legal mechanism of investment activities of agricultural enterprises to attract foreign capital investment. Restoration of the agricultural complex is possible through the use of a set of interrelated factors of financial, economic, production, technical, organisational and social nature, which are aimed at creating stabilising mechanisms, thereby improving the efficiency and competitiveness of agricultural enterprises [14].

Given the presented deliberations, the purpose of the study is to develop a scientific and methodological approach to the formation and selection of strategies to improve the agricultural potential of the country's regions. The paper is structured as follows. Firstly, the theoretical background of the study is presented. The theoretical deliberations concentrate on choosing a strategy for agricultural development. Secondly, the research methodology is presented. The next part of the paper highlights the results. Finally, the conclusions and limitations of the study are presented.

## 2. Literature Review

The issue of the agricultural resource potential development is becoming increasingly important for countries in the whole world. Cooperation between different countries is an important political issue related to European integration [15,16]. In this context, special attention should be paid to dynamic factors that create new opportunities and threats for the agricultural sector [17] such as climate change, environmental issues [18], introduction of innovations [19], investment attraction, as well as increased emphasis on CSR aspects [20], etc. These preconditions shape interest in a more thorough study of the formulation of a strategy for the development of agricultural resource potential.

Many scholars from different countries have been working on this topic. Each of them reveals its certain aspect that is most relevant for the particular country, given the geographical location and climatic conditions [21–23], or that is most dangerous for the ecology of the World [24], or that is connected to innovations which are the engine for the agricultural sector [25,26].

It should be emphasised that the resource potential of agro-industry directly depends on minimising the factors of negative impact on the development of agro-industrial enterprises that cause negative fluctuations in the agricultural sector [27,28]. One of the crucial aspects is ecology. Outdated agricultural technologies and techniques have a significant negative impact on the environment [29]. Due to the intensive farming and use of various plastic materials, agriculture contributes to the rapid growth of plastic production and consumption worldwide. Thus, it is necessary to conduct an environmental inventory by compiling an environmental rating of technologies. The results of research show that crop production technologies have a significant negative impact on the environment, climate risks and climate change, land fertility, etc. [30]. Under such conditions, it is necessary to select appropriate technologies adapted to environmental sensitivity. Due to climate change, special attention is paid to environmental management of the agri-food sector [31]. In particular, modern research, development and innovation (RDI) are aimed at transforming conventional agricultural production into a sustainable and environmentally-friendly industry. An important role for the creation of environmental effect is played by the potential of the bioprocessing system, which is based on the processing of waste into useful bioproducts, such as production of grapeseed oil [32]. This approach has a positive effect on the economic conditions of enterprises [33], as it not only eliminates waste, but also creates additional income.

Describing the factors influencing the agricultural resource potential, it can be noted that they are interdependent and complementary, because the establishment of the waste-free production requires the introduction of new technologies. Under such conditions, another key aspect for the development of agricultural resource potential is investment and innovation activities of agricultural organisations [34]. When revealing the importance of innovation and investment activities in the agro-industrial sector, it is necessary to take into account industry specifics. In particular, for successful implementation of innovations it is necessary to: (1) analyse the problems of innovation in agricultural organisations; (2) to develop criteria for innovation that can be used in the investment projects of the agricultural organisation; (3) to describe the system of innovations and the structure of innovation activities in the agricultural sector; (4) to describe barriers for innovation in agriculture [35]; (5) to choose measures to minimise the risks of leasing land and equipment [36], etc.

One should add that the development of the potential of agriculturedepends directly on the rational use of available natural resources [37,38]. That is why the selection of regional strategies for optimising land use is so important [39]. It should take into account the impact of topography, climate and social demographic factors on the ecosystem. Economic assessment and use of land resources of agricultural enterprises can serve as an information base for the categorization of fertile land. In turn, this can be used for the implementation of measures to optimise land use and for regulatory assessment of land [40]. In addition, due to recent advances in artificial intelligence it is possible to quantify phototype information about the scale of fertile land and predict their quality use [41]. It is also possible to study

the impact of environmental agriculture on soil structure and the hydraulic properties of agricultural systems [42]. With the help of an innovative tool for ecological assessment of agricultural production systems, one can calculate the index of ecosystem integrity [43], which can be helpful in selecting a strategy for development of the agricultural resource potential. Nanotechnology in agriculture should also be mentioned [44]. It promotes exploitation of nanofertilisers, nanopesticides, nanobiosensors and nanotilitary strategies for the restoration of contaminated soils. Researchers and farmers focus on new prospects for the development of farms [45,46], agri-food supply networks and rural development according to the new common international agricultural policy.

However, despite a significant amount of research, today there is no systematic approach relates to the factors and processes of the strategy selection to improve the agricultural potential of regions for developing the bio-economy [47]. Managerial decisions are often made intuitively or subjectively, which does not allow a comprehensive assessment of the state of and opportunities for regional agricultural development. This primarily affects the quality of decisions results—low profitability and production, inefficient use of resources, reduction of production, investment in unprofitable areas of agriculture, etc. The European Union supports agricultural policies, and in the context of the pandemic COVID-19 has initiated programs to support EU countries to develop agricultural production and facilitate the sale of agricultural products obtained by them [48]. All this necessitates further research and substantiation of scientific and methodological approaches to strategies selection to improve the agricultural potential of regions, which reflects the relevance of the chosen research topic.

Analysing the characteristics of the development of strategies for the agro-industrial sector [49,50], it can be noted that the authors consider, in particular, emphasizing organizational and managerial issues (e.g., the need for a leader with strong environmental analysis and strategic management skills as well as decision-making skills pertaining to the implementation of strategy). However, researchers do not pay enough attention to the development of a strategy for an agricultural sector in general, given its potential profitability and investments. Most of the studies on the formation of strategies for agricultural resource potential development lack a comprehensive approach, and do not take all factors of the internal and external environment into account. Moreover, one cannot recommend a strategy for the sector without taking into account the nature of the agricultural sector in that particular country. In other words, there is no doubt that two such different countries as—for example—the Netherlands and Ukraine will apply different strategies to their agricultural sectors. In the case of the latter, one observes a broad use of concentrated strategies and the creation of large agricultural holdings [4]. Also, the mergers and acquisitions of agricultural enterprises are used on a broad scale, which is in confirmed by general of this phenomenon in relation to Ukraine [51]. Moreover, in the case of the Ukrainian agricultural sector one also claims on sustainable development strategy; however, this notion is understood not only in relation to the agricultural sector but in a broader sense, i.e., it also relates to the rural territories [6].

The studies on the formation of strategies for the development of resource potential of the agricultural sector are generalised for countries. However, they don't take into account the level of resource development of regions. It can be significantly different for each of them, requiring separate strategies for each region and not a single common strategy for the whole country, It is also noteworthy that most studies are aimed at qualitative analysis of factors, but there is no detailed method of taking into account the importance of quantitative factors (resource potential of the region, product profitability, investment security of the region, etc.). Such preconditions have created the need to develop tools for management decisions making in the context of agricultural development based on the results of statistical analysis and construction of an appropriate scientific and methodological model. Given these facts, we formulate the following main hypothesis:

The choice of strategy of development of agro-branch of a region depends on three key parameters: (1) resource potential of the region; (2) product profitability; and (3) investment support of the region.

To confirm it, the task is to develop a methodology for choosing a strategy to improve the resource potential of the region depending on the group to which the region was previously assigned according to the level of the above parameters. At its formation it is necessary to answer the following questions:

Q1. What strategy to improve the resource potential of regions in the field of crop production should be chosen on the basis of comparing clusters of regions according to the level of resource potential of crop production and the level of its profitability?

Q2. What strategy to improve the resource potential of regions in the field of livestock should be chosen based on the comparison of clusters of regions by the level of development of the resource potential of livestock and the level of its profitability?

Q3. What investment strategy for agricultural development should be chosen on the basis of comparing clusters of regions according to the level of development potential of crop and livestock resources and the level of investment support for agriculture?

## 3. Materials and Methods

To substantiate the main hypothesis and form answers to the above questions, the research methodology used in this article consists of the following main stages:

Stage 1   Selection of indicators for clustering of regions according to the level of development potential of crop production and animal husbandry as well as the level of profitability of crop production and animal husbandry;

Stage 2   grouping of the country's regions according to the level of development of crop and livestock potential by cluster analysis using the method of K-means;

Stage 3   grouping of regions according to the level of profitability of crop and livestock products through cluster analysis using the predominance function;

Stage 4   grouping of the regions by the level of investment support of agriculture by the statistical method;

Stage 5   selection of agricultural development strategy of the region depending on the importance of its resource potential, profitability of products and investment support.

### 3.1. Method of Cluster Analysis

It is proposed to select strategies for improving the agricultural resource potential of the country's regions using a methodology that, unlike others, is based on comparing clusters of regions and groups formed by the profitability of crop and stockbreeding production and the investment level in agriculture. The advantage of the proposed method is that cluster analysis involves the multidimensionality of statistical research, which includes collecting data about sample objects and organising them into relatively homogeneous groups. Cluster analysis is a means of grouping multidimensional objects and differs from conventional grouping in that each cluster contains similar objects that differ sharply from objects of other clusters. Grouping of regions according to the profitability of agricultural products and the level of investment provision of agriculture will allow the collection of necessary, reasonable and timely information as well as to monitor the state of and opportunities for agricultural development of each region. This, in turn, will enable the selection of the optimal strategy for each region.

The abovementioned advantages of using cluster analysis explain its wide application in various fields: geography, biology, statistics, psychology, economics and other social sciences [52–54]. Cluster analysis is often used to group enterprises or regions according to a set of socio-economic indicators. A study of the literature showed that it was used by scientists to determine the clusters of small and medium-sized enterprises operating in the Slovak Republic and the use of marketing communication tools in the future [55], to identify groups of Russian regions with similar family problems [56], to assess the selected structural aspects of regional competitiveness and their role in identifying regional

imbalances at the level of self-governing regions of the Slovak Republic [57], to group the united territorial communities of Zaporizhzhia region with similar rates of economic development [58], etc.

In the study, the clustering of regions by the level of agricultural potential development using the method of K-means is conducted. The main task of the cluster analysis is the division of a large number of studied objects and features into homogeneous groups or clusters. Thus, the appropriate structure is identified and the problem of data grouping is solved [59,60]. The K-means algorithm solves the problem of the existence of assumptions for the number of clusters; herein, they should be as different as possible [61].

The object of clustering are the regions of Ukraine, which is described by the vector:

$$X = \{X_1, X_2, \ldots, X_n\}, \tag{1}$$

where $X_1, X_2, \ldots, X_n$—indicators that characterise the regions of Ukraine;

$n$—determines the dimension of the characteristic's space. The geometric proximity of two or more points in this space means that these points belong to the same cluster.

The proposed algorithm of cluster analysis is presented in Figure 1.

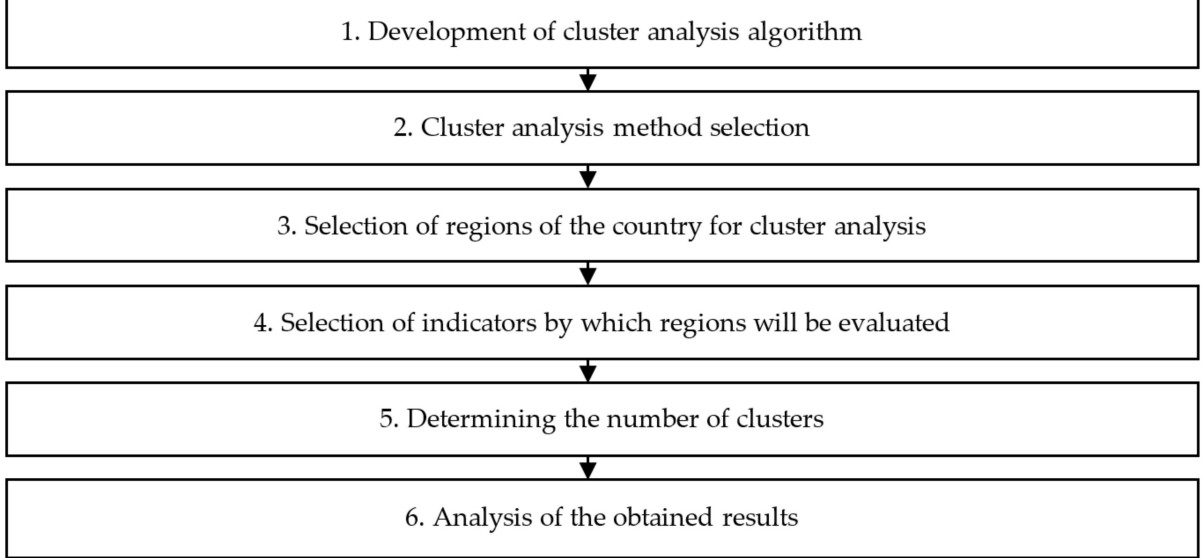

**Figure 1.** Algorithm for cluster analysis of the country's regions by the level of agricultural potential development. Source: own elaboration based on [62–64].

The whole algorithm consists of several stages. The first stage is preparation of the algorithm for cluster analysis, determination of features and methods of cluster analysis.

The second stage is selection of K-means clustering for regions according to the level of development of stockbreeding and crop production potential. When selecting the method of clustering, we focused primarily on its ease of use and informativeness compared to other methods. K-means clustering is more convenient for developing characteristic capacity graphs, because one can create several sets with different numbers of characteristic capacity graphs and comparing these sets with each other. In result, it allows the choosing of the optimal one.

In turn, the third stage is the selection of 24 regions of Ukraine (not including the Autonomous Republic of Crimea and cities of state importance). In the fourth stage the selection of indicators for cluster analysis is conducted. For each of two industries, the main indicators were selected, according to which the clustering was carried out. The fifth stage is the determination of the number of clusters. Based on the set of indicators, six clusters were formed for crop production, and four clusters for stockbreeding. The method of cluster analysis was applied using the software package STATISTICA 10 (StatSoft

Company, USA, 2300 East 14th Street, Tulsa, OK 74104). The last, sixth stage is analysis and visualisation of the obtained results, i.e., the graphic and tabular presentation of the results and determination of the position of each region in relation to the others.

### 3.2. Selection of Indicators for Regional Clustering

The most important aspect of clustering is the adequate selection of variables for clustering. Even one incorrectly selected variable can distort the results of clustering. That is why indicators that best describe the similarity of objects in terms of the most influential features were chosen. The set of indicators was formed taking into account the principles of representativeness (the most significant indicators affecting the level of agricultural potential were selected), information accessibility (availability of open access statistics) and reliability (indicators that adequately reflect the level of development of the agricultural potential). All the data used for our analysis are taken from the official website of the State Statistics Service of Ukraine [65] and its statistical collections [66,67]. The level of development of agricultural potential in the context of its two main branches, i.e., crop production and stockbreeding was assessed.

The list of indicators selected for cluster analysis of regions include:

(1) According to the level of development of crop potential:

- sown areas of crops [in thousands of hectares];
- indices of crop production of farms of all categories [%];
- crop production per capita [UAH] (1 UAH = 0.0306 EUR (on 3 April 2021));
- labour productivity at agricultural (crop) enterprises [UAH];
- gross harvest of cereals and legumes [in thousands of tons].

(2) According to the level of development of stockbreeding potential:

- volume of farm animals breeding [in thousands of tons];
- indices of stockbreeding production of farms of all categories [%];
- stockbreeding products per capita [UAH];
- labour productivity at agricultural (stockbreeding) enterprises [UAH];
- number of cattle [in thousands of heads].

Table 1 shows statistical data for the cluster analysis of regions according to the level of development of crop potential.

**Table 1.** Distribution of regions of Ukraine into clusters according to the level of development of crop potential, 2018.

| Regions of Ukraine | Symbols | Sown Areas of Crops, in Thousands of Hectares | Indices of Crop Production of Farms of All Categories, % | Crop Production Per Capita, UAH | Labour Productivity in Agricultural (Crop) Enterprises, UAH | Gross Harvest of Cereals and Legumes, in Thousands of Tons |
|---|---|---|---|---|---|---|
| Vinnytsia region | C1 | 1625 | 114.1 | 9920 | 331,070.3 | 5911.1 |
| Volyn region | C2 | 577 | 106.9 | 4249 | 359,635.6 | 1237.2 |
| Dnipropetrovsk region | C3 | 1953 | 105.3 | 3480 | 243,516.5 | 3487.5 |
| Donetsk region | C4 | 1004 | 85.9 | 1084 | 190,790.2 | 1344.4 |
| Zhytomyr region | C5 | 1042 | 115.3 | 6692 | 402,973.3 | 2424.1 |
| Transcarpathian region | C6 | 189 | 105.3 | 1749 | 145,932 | 375.9 |
| Zaporizhzhia region | C7 | 1672 | 83.1 | 3723 | 180,828.7 | 2233.3 |
| Ivano-Frankivsk region | C8 | 381 | 101.1 | 2358 | 304,553.4 | 804.5 |
| Kyiv region | C9 | 1191 | 129.8 | 2528 | 280,747.3 | 4081.5 |
| Kirovohrad region | C10 | 1703 | 125 | 11,105 | 279,710.9 | 3763.2 |
| Luhansk region | C11 | 825 | 108.8 | 1971 | 272,699.3 | 1159.4 |
| Lviv region | C12 | 675 | 104.3 | 2565 | 388,793.5 | 1440 |
| Mykolaiv region | C13 | 1565 | 108.6 | 6879 | 252,300.7 | 2673.4 |
| Odessa region | C14 | 1860 | 102.4 | 4267 | 241,534.6 | 4319.9 |

**Table 1.** *Cont.*

| Regions of Ukraine | Symbols | Sown Areas of Crops, in Thousands of Hectares | Indices of Crop Production of Farms of All Categories, % | Crop Production Per Capita, UAH | Labour Productivity in Agricultural (Crop) Enterprises, UAH | Gross Harvest of Cereals and Legumes, in Thousands of Tons |
|---|---|---|---|---|---|---|
| Poltava region | C15 | 1719 | 133.0 1 | 10,042 | 328,358.6 | 6341.8 |
| Rivne region | C16 | 575 | 104.8 | 4325 | 386,665.1 | 1259.5 |
| Sumy region | C17 | 1162 | 113.7 | 8529 | 491,638.8 | 4470.1 |
| Ternopil region | C18 | 839 | 104.6 | 7186 | 419,574.6 | 2631.9 |
| Kharkiv region | C19 | 1793 | 107.1 | 4445 | 318,703.2 | 3829.2 |
| Kherson region | C20 | 1396 | 100.8 | 8817 | 253,383.5 | 2267.7 |
| Khmelnytsky region | C21 | 1186 | 104.7 | 8709 | 420,937.3 | 3861 |
| Cherkasy region | C22 | 1188 | 138.2 | 8600 | 305,361.7 | 4644 |
| Chernivtsi region | C23 | 307 | 108.4 | 3492 | 245,523.7 | 586.4 |
| Chernihiv region | C24 | 1272 | 114.1 | 9905 | 377,357.8 | 4909.5 |

Source: compiled by the authors based on [66].

Table 2 shows statistical data for the cluster analysis of regions according to the level of development of stockbreeding potential.

**Table 2.** Distribution of regions of Ukraine into clusters according to the level of development of stockbreeding potential, 2018.

| Regions of Ukraine | Symbols | Volume of Farm Animals Breeding, in Thousands of Tons | Indices of Stockbreeding Production of Farms of All Categories, % | Stockbreeding Products Per Capita, UAH | Labour Productivity in Agricultural (Stockbreeding) Enterprises, UAH | Number of Cattle, in Thousands of Heads |
|---|---|---|---|---|---|---|
| Vinnytsia region | C1 | 476.2 | 103.5 | 4486 | 671,800.6 | 239.4 |
| Volyn region | C2 | 153.3 | 97.6 | 2588 | 435,109.9 | 130.3 |
| Dnipropetrovsk region | C3 | 326.7 | 96.7 | 1387 | 547,424.7 | 122.1 |
| Donetsk region | C4 | 122.4 | 100.7 | 579 | 378,923.2 | 59.7 |
| Zhytomyr region | C5 | 86.6 | 103.3 | 2387 | 180,450.2 | 189.4 |
| Transcarpathian region | C6 | 83.3 | 108.8 | 1671 | 123,612.4 | 122.9 |
| Zaporizhzhia region | C7 | 66.9 | 95.7 | 1072 | 235,448.9 | 91.5 |
| Ivano-Frankivsk region | C8 | 128.4 | 101.6 | 2083 | 485,685.6 | 136.2 |
| Kyiv region | C9 | 375.4 | 114.3 | 1390 | 427,957.7 | 117.1 |
| Kirovohrad region | C10 | 69.3 | 101.3 | 2044 | 187,174 | 89.7 |
| Luhansk region | C11 | 22.8 | 112 | 319 | 99,813.7 | 52.4 |
| Lviv region | C12 | 192.6 | 102.8 | 1469 | 491,214.6 | 170.9 |
| Mykolaiv region | C13 | 41.1 | 94.6 | 1361 | 184,810 | 98.5 |
| Odessa region | C14 | 59 | 94.6 | 748 | 136,343.7 | 154.9 |
| Poltava region | C15 | 97.5 | 97.9 | 2570 | 206,068.1 | 231.3 |
| Rivne region | C16 | 81.9 | 97.8 | 1920 | 287,702.3 | 118.5 |
| Sumy region | C17 | 65.8 | 102.6 | 1916 | 159,967.5 | 146.3 |
| Ternopil region | C18 | 74.7 | 101.6 | 2190 | 387,897.8 | 138.7 |
| Kharkiv region | C19 | 118.6 | 102 | 1123 | 243,943.3 | 180.8 |
| Kherson region | C20 | 66.3 | 99.2 | 1970 | 358,611.1 | 96 |
| Khmelnytsky region | C21 | 99.6 | 96.4 | 2654 | 271,622.9 | 230.2 |
| Cherkasy region | C22 | 449.2 | 102.4 | 4821 | 491,745.9 | 161 |
| Chernivtsi region | C23 | 62.4 | 99.6 | 1754 | 301,821.2 | 81.5 |
| Chernihiv region | C24 | 48.2 | 98.8 | 2078 | 171,315.3 | 173.6 |

Source: compiled by the authors based on [68].

### 3.3. Methodology of Grouping of Regions by the Level of Profitability of Crop and Stockbreeding Products

Classification of multidimensional observations or objects is based on the definition of the distance between the studied objects. The most common is the Euclidean distance [69,70]. This distance between the objects $j$ and $k$ is a geometric distance in multidimensional space and is calculated by the formula:

$$d_{jk} = \sqrt{\sum_{i=1}^{m} (x_{ij} - x_{ik})^2} \tag{2}$$

where $d_{jk}$—the distance between the objects $j$ and $k$; $x_{ij}$—the value of the $j$-th object on the $i$-th indicator; $x_{ik}$—the value of the $k$-th object on the $i$-th indicator.

The coordinates of the point $P_0 = (z_{01}, z_{02}, \ldots, z_{0n})$ are determined, which is called the standard (the highest value for each of the features when this feature is a stimulator, and the lowest value when the feature is a destimulator). Based on the results, the distance from the point $P_i$ to the point $P_0$ can be calculated by the formula:

$$d_{j0} = \sqrt{\frac{1}{m} \sum_{j=1}^{n} (z_{ij} - z_{0j})^2} \tag{3}$$

Calculation of the advantage function of the profitability level of agricultural products of the $j$-th region $f(x_j)$ is carried out according to the formula:

$$f(x_j) = 1 - \frac{d_{j0}}{d_o} \tag{4}$$

$$d_0 = \bar{d} + aS_d \tag{5}$$

$$\bar{d} = \frac{1}{m} \sum_{i=1}^{n} d_{i0} \tag{6}$$

$$S_d = \sqrt{\frac{1}{m} \sum_{i=1}^{n} \left(d_{i0} - \bar{d}\right)^2} \tag{7}$$

where $d_{i0}$—the distance from the point $P_i$ to the point $P_0$, $a$—some positive number that is chosen so that all values of the function $f(x_j)$ are between zero and one (in this case $a = 3$). According to this model of calculating the advantage function (the integrated level of profitability of agricultural products is ideally equal to 1), the closer the value of the advantage function $f(x_j)$ of the $j$-th region is to 1, the higher is the integrated level of profitability of agricultural products within it. Table 3 presents statistical data for the grouping of regions by the level of profitability of crop production of agricultural enterprises in 2018, while Table 4 presents statistical data for the grouping of regions by the level of profitability of livestock products of agricultural enterprises in 2018.

**Table 3.** The level of profitability of crop production of agricultural enterprises by region in 2018.

| Regions of Ukraine | The Level of Profitability of Crop Production, % | | | | | |
|---|---|---|---|---|---|---|
| | Cereals and Legumes | Sunflower | Sugar (Factory) Beets | Vegetables | Potatoes | Fruits and Berries |
| | $x_1$ | $x_2$ | $x_3$ | $x_4$ | $x_5$ | $x_6$ |
| Vinnytsia region | 28.1 | 37.4 | −19.6 | −13.0 | −2.2 | 6.0 |
| Volyn region | 27.6 | 20.6 | −6.3 | 31.2 | −2.3 | 179.6 |
| Dnipropetrovsk region | 24.6 | 37.8 | 0.3 | 41.4 | 1.2 | 7.1 |
| Donetsk region | 23.8 | 32.6 | 0.0 | 27.9 | −4.2 | 4.8 |
| Zhytomyr region | 23.1 | 22.4 | −16.2 | 10.0 | 29.0 | 70.3 |
| Transcarpathian region | 18.3 | 3.7 | 0.0 | −22.8 | −19.5 | 16.5 |

**Table 3.** *Cont.*

| Regions of Ukraine | The Level of Profitability of Crop Production, % | | | | | |
| --- | --- | --- | --- | --- | --- | --- |
| | Cereals and Legumes | Sunflower | Sugar (Factory) Beets | Vegetables | Potatoes | Fruits and Berries |
| | $x_1$ | $x_2$ | $x_3$ | $x_4$ | $x_5$ | $x_6$ |
| Zaporizhzhia region | 21.0 | 27.7 | 0.0 | 29.0 | 10.9 | −27.3 |
| Ivano-Frankivsk region | 6.2 | 4.0 | −9.0 | −7.5 | −2.8 | 50.8 |
| Kyiv region | 24.2 | 31.1 | −24.4 | 6.6 | 6.7 | −4.3 |
| Kirovohrad region | 20.5 | 30.6 | −39.7 | −60.8 | −7.1 | −27.2 |
| Luhansk region | 22.4 | 30.5 | 4.2 | −59.4 | 0.0 | −9.7 |
| Lviv region | 17.5 | 12.4 | 8.1 | 4.5 | 23.5 | −1.5 |
| Mykolaiv region | 32.2 | 38.0 | 0.0 | 15.6 | −21.7 | −29.8 |
| Odessa region | 28.1 | 30.8 | 0.0 | 28.1 | −3.9 | −2.2 |
| Poltava region | 23.9 | 31.5 | 3.2 | 5.4 | −43.4 | −1.7 |
| Rivne region | 17.9 | 19.5 | −49.6 | 4.0 | 53.6 | 16.7 |
| Sumy region | 28.6 | 35.8 | −6.7 | 6.8 | 12.0 | −6.7 |
| Ternopil region | 27.8 | 28.5 | −20.3 | −1.3 | −53.5 | 21.2 |
| Kharkiv region | 16.7 | 32.5 | −16.8 | −0.3 | 19.8 | −9.0 |
| Kherson region | 28.0 | 26.2 | 0.0 | 0.8 | 2.5 | −17.5 |
| Khmelnytsky region | 32.0 | 32.6 | −10.1 | 24.0 | 51.8 | 23.0 |
| Cherkasy region | 32.9 | 45.4 | −40.0 | 26.3 | 16.6 | −49.5 |
| Chernivtsi region | 9.7 | 34.1 | 0.0 | 181.3 | 50.9 | 6.9 |
| Chernihiv region | 17.9 | 28.1 | −19.8 | 21.0 | 20.3 | 38.5 |

Source: compiled by the authors based on [66].

**Table 4.** The level of profitability of livestock products of agricultural enterprises by region in 2018.

| Regions of Ukraine | The Level of Profitability of Stockbreeding, % | | | | | |
| --- | --- | --- | --- | --- | --- | --- |
| | Milk | Cattle for Meat | Pigs for Meat | Sheep and Goats for Meat | Poultry for Meat | Eggs |
| | $x_1$ | $x_2$ | $x_3$ | $x_4$ | $x_5$ | $x_6$ |
| Vinnytsia region | 17.9 | −7.7 | −4.9 | −31.3 | −8.7 | 7.5 |
| Volyn region | 30.6 | −0.8 | 14.7 | 31.6 | −8.7 | −96.8 |
| Dnipropetrovsk region | 24.4 | −14.3 | 8.6 | 11.4 | −22.1 | 13.8 |
| Donetsk region | 10.4 | −22.3 | −7.3 | −25.3 | −26.4 | 14.1 |
| Zhytomyr region | 13.7 | −25.4 | −4.2 | −65.1 | −7.3 | 22.1 |
| Transcarpathian region | −22.8 | −36.0 | −1.0 | −9.6 | 30.6 | −6.4 |
| Zaporizhzhia region | 6.0 | −36.5 | −12.7 | −29.9 | −6.6 | 14.4 |
| Ivano-Frankivsk region | 23.3 | −8.7 | 18.8 | 37.6 | 0.8 | −24.0 |
| Kyiv region | 8.7 | −16.6 | 15.4 | 1.3 | 18.6 | 14.2 |
| Kirovohrad region | 0.1 | −26.6 | −0.4 | −15.3 | 15.5 | −20.5 |
| Luhansk region | 6.1 | −30.4 | −27.9 | −17.9 | −22.5 | 39.6 |
| Lviv region | 5.9 | −7.5 | 3.2 | −3.0 | 4.4 | 13.4 |
| Mykolaiv region | 28.7 | −8.4 | 4.1 | −19.7 | −59.5 | −6.0 |
| Odessa region | 7.4 | −23.8 | −3.3 | −26.4 | −11.2 | −11.2 |
| Poltava region | 17.2 | −27.5 | −10.8 | −48.0 | −10.4 | 23.4 |
| Rivne region | 11.3 | −7.4 | −0.1 | 3.5 | 9.7 | 37.2 |
| Sumy region | 20.1 | −11.5 | 0.7 | −17.7 | −21.4 | −8.4 |
| Ternopil region | 12.6 | −10.4 | 21.8 | 22.7 | −21.7 | 32.4 |

**Table 4.** *Cont.*

| Regions of Ukraine | The Level of Profitability of Stockbreeding, % | | | | | |
|---|---|---|---|---|---|---|
| | Milk | Cattle for Meat | Pigs for Meat | Sheep and Goats for Meat | Poultry for Meat | Eggs |
| | $x_1$ | $x_2$ | $x_3$ | $x_4$ | $x_5$ | $x_6$ |
| Kharkiv region | 14.7 | −18.8 | 4.3 | −47.2 | 17.2 | −22.0 |
| Kherson region | 13.6 | −15.1 | 8.9 | −51.5 | −72.5 | 0.3 |
| Khmelnytsky region | 23.8 | −19.1 | 21.1 | −7.6 | −8.3 | 15.5 |
| Cherkasy region | 17.0 | −17.2 | 3.1 | −4.2 | −17.2 | −76.1 |
| Chernivtsi region | 15.7 | −19.2 | −13.5 | −22.3 | −6.2 | 4.6 |
| Chernihiv region | 21.8 | −20.5 | −1.8 | −55.8 | −43.3 | −0.8 |

Source: compiled by the authors based on [67].

### 3.4. Approach to the Choice of Investment Strategy

The deliberations presented allow us to propose a scientific and methodological approach to the formation and selection of strategies to improve the agricultural potential of regions (Figure 2).

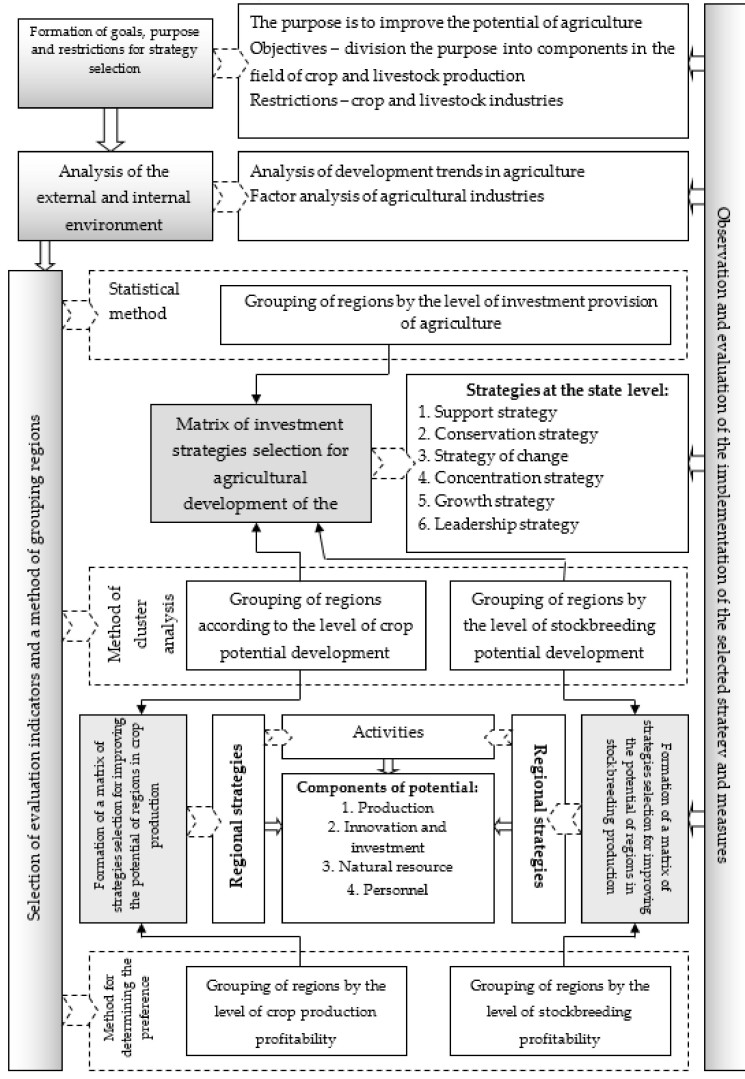

**Figure 2.** Scientific and methodological approach to the formation and selection of strategies to improve the agricultural potential of regions.

The matrix method of strategy selection has been widely used in strategic management since the 1960s. Its advantage is that it allows the choice to be made when concurrently taking several parameters into account. The most common are two-factor matrices: the Boston Consulting Group (BCG) matrix to analyze growth rates (growth) and market share [71], the General Electric McKinsey matrix to analyze market attractiveness and competitiveness [72], the Arthur D. Little matrix (ADL/LC) for the analysis of the life cycle of the industry and the relative state of the market [73], the "Product–market" matrix (Ansoff matrix) for the choice of strategy for markets and goods [74], etc. The advantages of the matrix method are simplicity, clarity, and the possible dynamic mode, while a disadvantage is that only two parameters are taken into account. In order to take the three parameters that are investigated in the study into account, a 3D matrix is proposed. Based on the cluster analysis of the regions of Ukraine by the level of development of crop and stockbreeding potential and their grouping by the level of investment provision of agriculture, a 3D matrix for the selection of an investment strategy for agricultural development in Ukraine was built (Figure 3). The X axis shows clusters of regions according to the level of development of crop potential, the Y axis shows clusters of regions according to the level of development of stockbreeding potential, and the Z-axis shows groups of regions according to the level of investment provision for agriculture.

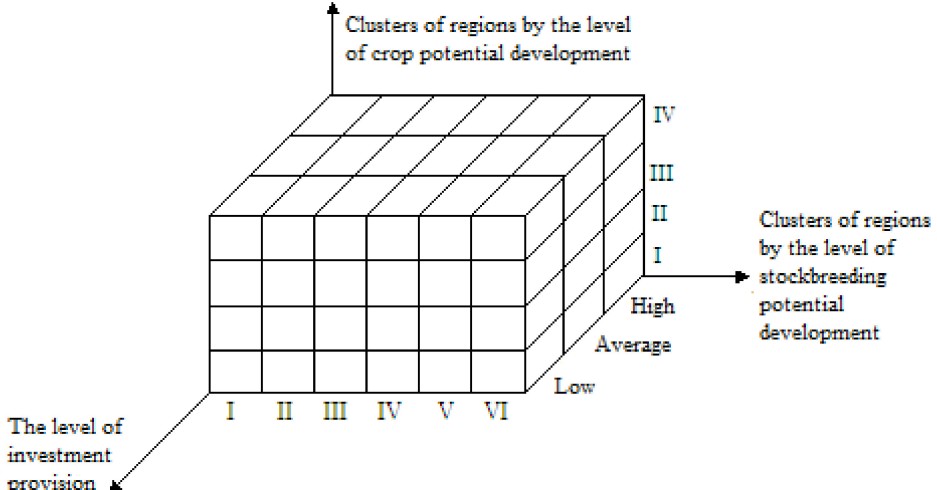

**Figure 3.** 3D matrix for the selection of an investment strategy for agricultural development in Ukraine.

## 4. Results

### 4.1. Development of Crop Potential

Figure 4 presents the dendrogram of the regions of Ukraine by the development of crop potential according to the 2018 data.

Table 5 shows the division of the regions of Ukraine into six clusters according to the level of development of crop potential based on 2018 data.

The first cluster includes mainly the most developed regions of Ukraine (four out of 24 studied). Their position is characterised by fairly high production indices, a significant sown area of crops, labour productivity, the volume of crop production per capita and the gross harvest of cereals and legumes. In turn, the second cluster consists of two regions (Mykolayiv and Kherson regions). They are close enough to the values of indicators that characterise the regions of the first cluster. Like the second cluster, the third cluster also includes two regions. They have an obvious lag in the level of development of crop potential compared to the first and second clusters. For example, crop production per capita in this cluster is UAH 2403.5, while in the second the average value of the same indicator is UAH 7848. The fourth cluster consists of four regions, but it should be noted that Dnipropetrovsk region, which is part of this group, has the largest number of sown areas of crops. The fifth cluster is the largest and consists of seven regions. They are united

by rather low sown areas and average productivity compared to other regions. Finally, the sixth cluster has the highest values of all indicators of development of crop potential. For results analysis, the average values (according to the initial actual data) of each indicator for the formed clusters were calculated (Table 6).

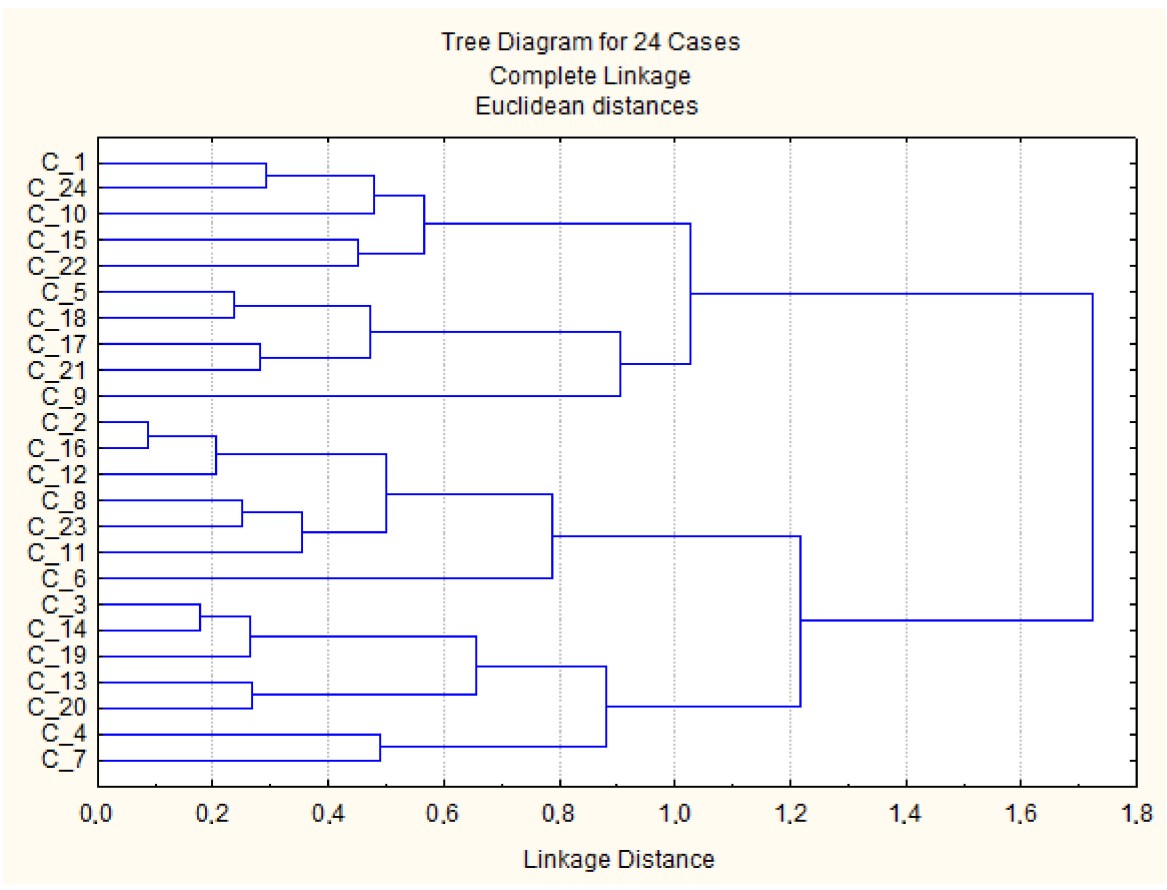

**Figure 4.** The dendrogram of the regions of Ukraine by the level of development of crop potential, 2018. Source: own elaboration based on [66].

**Table 5.** Division of the regions of Ukraine into clusters according to the level of development of crop potential, 2018.

| I Cluster | II Cluster | III Cluster |
|---|---|---|
| Zhytomyr region (C5) Sumy region (C17) Ternopil region (C18) Khmelnytsky region (C21) | Mykolaiv region (C13) Kherson region (C20) | Donetsk region (C4) Zaporizhzhia region (C7) |
| **IV Cluster** | **V Cluster** | **VI Cluster** |
| Dnipropetrovsk region (C3) Kyiv region (C9) Odessa region (C14) Kharkiv region (C19) | Volyn region (C2) Transcarpathian region (C6) Ivano-Frankivsk region (C8) Luhansk region (C11) Lviv region (C12) Rivne region (C16) Chernivtsi region (C23) | Vinnytsia region (C1) Kirovohrad region (C10) Poltava region (C15) Cherkasy region (C22) Chernihiv region (C24) |

*4.2. Development of Stockbreeding Potential*

Figure 5 presents the dendrogram of the regions of Ukraine by the development of stockbreeding potential according to the 2018 data.

**Table 6.** Average values of the studied indicators for the formed clusters of regions of Ukraine according to the level of development of crop potential, 2018. Source: own elaboration based on [67].

| Cluster | Number of Regions | Sown Area of Agricultural Crops, in Thousands of Hectares | Crop Production Indices of Farms of All Categories, % | Crop Production Per Capita, UAH | Labour Productivity at Agricultural Enterprises, UAH | Gross Harvest of Cereals and Legumes, in Thousands of Tons |
|---|---|---|---|---|---|---|
| I | 4 | 1057.25 | 109.575 | 7779 | 433781 | 3346.775 |
| II | 2 | 1480.5 | 104.7 | 7848 | 252,842.1 | 2470.55 |
| III | 2 | 1338 | 84.5 | 2403.5 | 185,809.45 | 1788.85 |
| IV | 4 | 1699.25 | 111.15 | 3680 | 271,125.4 | 3929.525 |
| V | 7 | 504.14 | 105.65 | 2958.42 | 300,543.22 | 980.41 |
| VI | 5 | 1501.4 | 122.85 | 9914.4 | 324,371.86 | 5113.92 |

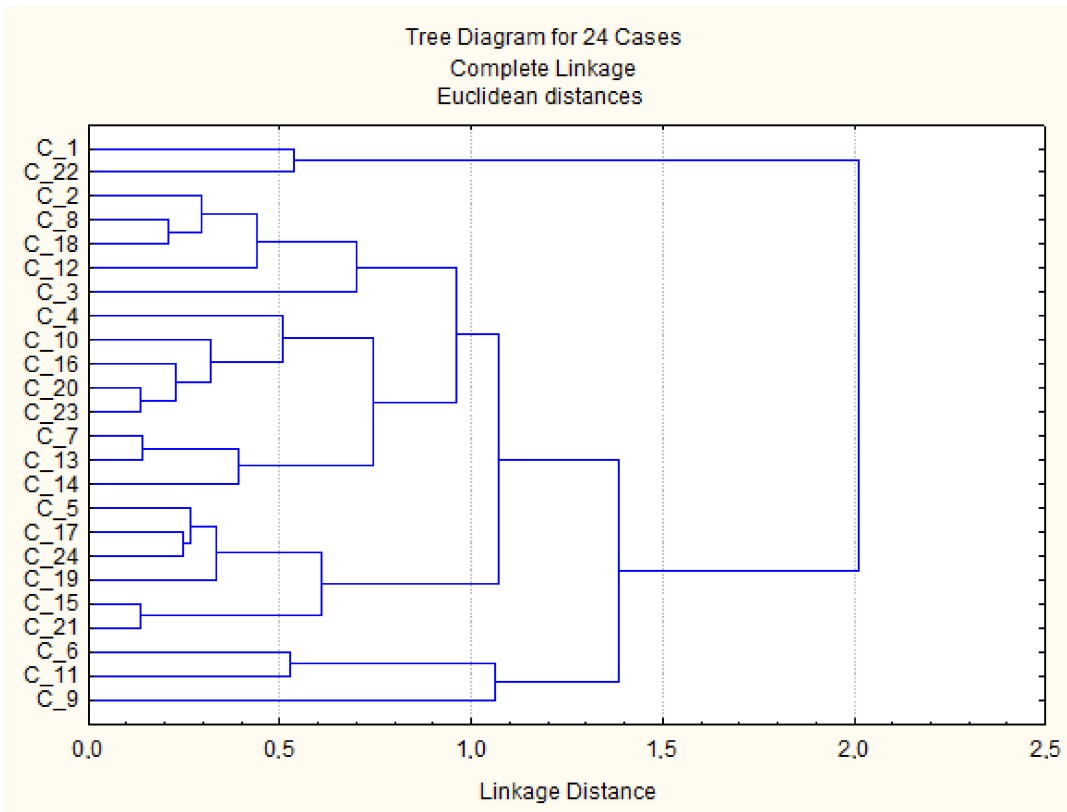

**Figure 5.** The dendrogram of the regions of Ukraine by the level of development of stockbreeding potential, 2018. Source: own elaboration based on [66].

Table 7 shows the division of regions of Ukraine into clusters according to the level of development of stockbreeding potential based on 2018 data.

Four clusters were identified according to the level of development of stockbreeding potential. The first cluster includes Volyn, Dnipropetrovsk, Ivano-Frankivsk, Kyiv, Lviv, and Ternopil regions. It is characterised by high productivity of agricultural enterprises. In turn, the second cluster is formed by seven regions with an average productivity and a positive trend towards capacity-building. The next, third cluster is larger and consists of 9 regions. This cluster is characterised by stagnation and a low development of stockbreeding. The last cluster includes the Vinnytsia and Cherkasy regions. The Vinnytsia region is a leader in the breeding of farm animals, the number of cattle and the labour productivity in agricultural enterprises, not only in this cluster but also in Ukraine as a whole. As for the Cherkasy region, it leads all regions in production per capita.

**Table 7.** Division of regions of Ukraine into clusters according to the level of development of stockbreeding potential, 2018.

| I Cluster | II Cluster |
|---|---|
| Volyn region (C2) Dnipropetrovsk region (C3) Ivano-Frankivsk region (C8) Kyiv region (C9) Lviv region (C12) Ternopil region (C18) | Zhytomyr region (C5) Odessa region (C14) Poltava region (C15) Sumy region (C17) Kharkiv region (C19) Khmelnytsky region (C21) Chernihiv region (C24) |
| **III Cluster** | **IV Cluster** |
| Donetsk region (C4) Transcarpathian region (C6) Zaporizhzhia region (C7) Kirovohrad region (C10) Luhansk region (C11) Mykolaiv region (C13) Rivne region (C16) Kherson region (C20) Chernivtsi region (C23) | Vinnytsia region (C1) Cherkasy region (C22) |

Source: own elaboration based on [68].

For results analysis, the average values (according to the initial actual data) of each indicator for the formed clusters were calculated (Table 8).

**Table 8.** Average values of the studied indicators for the formed clusters of regions of Ukraine according to the level of development of stockbreeding potential, 2018.

| Cluster | Number of Regions | Volume of Farm Animals Breeding, Thousands of Tons | Stockbreeding Production Indices of Farms of All Categories, % | Stockbreeding Products Per Capita, UAH | Labour Productivity at Agricultural Enterprises, UAH | Number of Cattle, Thousands of Heads |
|---|---|---|---|---|---|---|
| I | 6 | 208.52 | 102.43 | 1851.17 | 462,548.38 | 135.88 |
| II | 7 | 82.19 | 99.37 | 1925.14 | 195,673.00 | 186.64 |
| III | 9 | 68.49 | 101.08 | 1410.00 | 239,768.53 | 90.08 |
| IV | 2 | 462.70 | 102.95 | 4653.50 | 581,773.25 | 200.20 |

Source: own elaboration based on [68].

Thus, the cluster analysis made it possible to distinguish groups of regions of Ukraine according to the level of development of crop and stockbreeding production. The result of qualitative clustering is the clear differentiation between the obtained groups and similarities within the groups.

*4.3. Level of Profitability of Crop and Stockbreeding Products*

Using multidimensional cluster analysis, the advantage functions of the profitability level of crop and stockbreeding production in agriculture were calculated. They describe the ordered distance to the calculated ideal point of *n*-dimensional space, which is characterised by the highest values among stimulators and the lowest among destimulators. For calculations, statistical data of the State Statistics Service of Ukraine [65] characterising the level of profitability of crop and stockbreeding products of agricultural enterprises by region in 2018 were used. It is determined that the ideal point of profitability of crop production is:

$$P_0 = (32.9; 45.4; 8.1; 181.3; 53.6; 179.6),$$

while the ideal point of profitability of stockbreeding production is:

$$P_0 = (30.6; -0.8; 21.8; 37.6; 30.6; 39.6)$$

Three groups of regions by the value of the advantage function of the profitability level of crop and stockbreeding products in agriculture in 2018 were distinguished.

The range is found by the formula:

$$D = \frac{x_{\max} - x_{\min}}{n}, \tag{8}$$

where *n*—groups quantity, $X_{max}$, $X_{min}$—the maximum and minimum values of the advantage function, respectively.

For the level of profitability of crop production:

$$D = \frac{0.55 - 0.09}{3} = 0.15.$$

Intervals of groups of regions of Ukraine by the value of the advantage function $f(x_j)$ are:

Group I (low level of product profitability)—0.09–0.24;
Group II (average level of product profitability)—0.25–0.40;
Group III (high level of product profitability)—0.41–0.55.

Group I includes six regions of Ukraine with a low level of profitability of crop production. They are characterised by irrational use of resources, possible shortcomings in the organisation and management of activities, and low prospects for investment and new technologies adaptation.

Group II is the largest and includes 15 regions of Ukraine. It is characterised by the average level of profitability of crop production among the regions. These regions are a good springboard for the active implementation of new tillage technologies, the use of the latest equipment and methods of raw material processing.

Group III are leaders with high results, showing the ratio between available and used resources. This group includes three regions of Ukraine. The natural and economic conditions of these areas are favourable for doing business in the agricultural sector. The regions have significant land resources and investment activity.

For the level of profitability of stockbreeding products:

$$D = \frac{0.72 - 0.18}{3} = 0.18.$$

Intervals of groups of regions of Ukraine by the value of the advantage function $f(x_j)$ are:

Group I (low level of product profitability)—0.18–0.36;
Group II (average level of product profitability)—0.37–0.54;
Group III (high level of product profitability)—0.55–0.72.

Group I (seven regions) is characterised by a low level of profitability and strong stagnation in the industry. Each of the regions has negative profitability of a certain type of stockbreeding products: milk; cattle for meat; pigs for meat; sheep and goats for meat; poultry for meat; eggs, or even several types together. For these regions, it is necessary to create an effective economic mechanism, whose basis should be a combination of state regulation and self-regulation, the adaptation of new technologies, and a mixture of pricing, credit and tax policies. The second group is the largest and includes 11 regions with sufficiently developed stockbreeding, in the form mainly of poultry and pigs. This group is a good basis for investment, as the regions have a high resource potential and a sufficient raw material supply. The last group includes the regions of Dnipropetrovsk, Khmelnytsky, Lviv, Ternopil, Kyiv and Rivne.These regions are leaders among the studied regions and are characterised by the highest profitability of stockbreeding products.

The obtained values of the distance to the ideal point, the function of the advantage of the profitability level of crop and stockbreeding products, and the grouping of regions of Ukraine are given in Table 9.

**Table 9.** Grouping of regions of Ukraine by the value of the advantage function of the profitability level of crop and stockbreeding production, 2018.

| Regions of Ukraine | Distance to the Ideal Point, $d_{jo}$ | The Value of the Advantage Function, $f(x_j)$ | Integral Level of Product Profitability | Regions of Ukraine | Distance to the Ideal Point, $d_{jo}$ | The Value of the Advantage Function, $f(x_j)$ | Integral Level of Product Profitability |
|---|---|---|---|---|---|---|---|
| Crop production | | | | Stockbreeding production | | | |
| Kirovograd | 67.00 | 0.09 | Low level | Kherson | 29.42 | 0.18 | Low level |
| Luhansk | 63.57 | 0.14 | | Volyn | 29.04 | 0.19 | |
| Cherkasy | 57.81 | 0.22 | | Cherkasy | 27.55 | 0.23 | |
| Mykolayiv | 56.68 | 0.23 | | Chernihiv | 26.49 | 0.26 | |
| Transcarpathian | 56.13 | 0.24 | | Zhytomyr | 24.03 | 0.33 | |
| Kherson | 55.71 | 0.24 | | Mykolayiv | 24.03 | 0.33 | |
| Poltava | 55.35 | 0.25 | Average level | Kharkiv | 22.41 | 0.36 | Average level |
| Vinnytsia | 54.72 | 0.26 | | Poltava | 21.63 | 0.40 | |
| Ternopil | 54.40 | 0.26 | | Odessa | 20.54 | 0.43 | |
| Kharkiv | 54.29 | 0.26 | | Luhansk | 20.25 | 0.43 | |
| Zaporizhzhia | 53.36 | 0.28 | | Zaporizhzhia | 20.05 | 0.44 | |
| Kyiv | 53.18 | 0.28 | | Donetsk | 19.97 | 0.44 | |
| Sumy | 52.92 | 0.28 | | Transcarpathian | 19.32 | 0.46 | |
| Lviv | 52.55 | 0.29 | | Sumy | 19.08 | 0.47 | |
| Rivne | 50.91 | 0.31 | | Kirovograd | 19.07 | 0.47 | |
| Odessa | 50.05 | 0.32 | | Vinnytsia | 18.53 | 0.48 | |
| Ivano-Frankivsk | 49.22 | 0.33 | | Chernivtsi | 18.23 | 0.49 | |
| Donetsk | 49.05 | 0.34 | | Ivano-Frankivsk | 14.52 | 0.54 | |
| Dnipropetrovsk | 46.66 | 0.37 | High level | Dnipropetrovsk | 13.73 | 0.62 | High level |
| Khmelnytsky | 45.54 | 0.38 | | Khmelnytsky | 13.72 | 0.62 | |
| Chernihiv | 44.73 | 0.40 | | Lviv | 12.95 | 0.64 | |
| Zhytomyr | 42.38 | 0.43 | | Ternopil | 11.95 | 0.67 | |
| Chernivtsi | 35.69 | 0.52 | | Kyiv | 10.95 | 0.69 | |
| Volyn | 33.23 | 0.55 | | Rivne | 10.21 | 0.72 | |

In summation, it is worth noting that in order to increase the efficiency of agricultural enterprises, government support is needed in terms of improving the production capacity of the industry. The presented indicators reflect the positive trends in certain areas of agriculture and the potential for the development of currently unprofitable areas of agriculture.

### 4.4. The Level of Investment Provision in Agriculture

The creation of a favourable investment climate and increasing the level of investment are prerequisites for the restoration of Ukrainian agriculture. Groups of regions were distinguished according to the volume of capital investment in agriculture. The range is:

$$D = \frac{x_{\max} - x_{\min}}{n} = \frac{45.7 - 3.3}{3} = 14.14\%$$

As a result, three groups were formed (Table 10):
Group I (low level of investment provision)—3.3–17.4%;
Group II (average level of investment provision)—17.5–31.6%;
Group III (high level of investment provision)—31.7–45.7%.

**Table 10.** Grouping of regions of Ukraine by the level of investment provision of agriculture in 2018.

| Regions with a Low Level of Investment Provision | Regions with an Average Level of Investment Provision | Regions with a High Level of Investment Provision |
|---|---|---|
| Transcarpathian (3.3 *) Donetsk (4.4) Dnipropetrovsk (5.2) Lviv (5.3) Chernivtsi (11.9) Ivano-Frankivsk (12.3) Kharkiv (12.7) Zaporizhzhia (13.2) Odessa (14.0) Rivne (16.5) Kyiv (16.6) | Volyn (17.5) Poltava (22) Mykolayiv (23.6) Vinnytsia (27.3) Zhytomyr (27.3) Khmelnytsky (28.1) Kherson (30.5) | Ternopil (33.3) Cherkasy (35.1) Sumy (35.3) Luhansk (36.4) Chernihiv (44.5) Kirovograd (45.7) |

* Capital investments (used) in agriculture, % of the total volume in the region. Source: own elaboration based on [65].

As a result, 11 regions out of 24 are characterised by a low level of investment provision. Investments are mainly made at the expense of household or business owners. At the initial stage, they need government assistance to improve their condition for further attracting foreign investors. The next group includes the regions that are characterised by an average level of investment. Attracting new investments will contribute to their development, to the rational and efficient use of funds, and to improvements inthe quality of manufactured products. Lastly, six regions are classified as regions with a high level of investment provision. They are the basis for attracting foreign capital, expanding the range of products and the strengthening of position on the world agro-industrial market.

## 5. Discussion

Its central part is a matrix allowing choice of the strategy for a particular region based on a comparison of several indicators, in particular, cluster analysis and the level of investment security (or profitability level). The advantage of this approach is the ability to clearly choose a strategy, consider any region and change both parameters and regions in the matrix. This matrix is presented in Figure 6. The X axis represents clusters of regions of the level of crop potential development, whilst the Y axis represents clusters of regions by the level of profitability of crop production.

| | | VI cluster | V cluster | IV cluster | III cluster | II cluster | I cluster |
|---|---|---|---|---|---|---|---|
| Clusters of regions by the profitability level of crop production | Group III (high level) | Strategy 2 | Strategy 3 | Strategy 3 | Strategy 4 | Strategy 4 | Strategy 4 |
| | Group II (inter-mediate level) | Strategy 1 | Strategy 2 | Strategy 2 | Strategy 3 | Strategy 3 | Strategy 4 |
| | Group I (low level) | Strategy 1 | Strategy 1 | Strategy 2 | Strategy 2 | Strategy 3 | Strategy 3 |

Clusters of regions by the level of crop potential development

**Figure 6.** Matrix of strategies selection for improving the potential of regions in crop production. Source: own elaboration.

Based on the comparison of the formed clusters of regions using cluster analysis and groups of regions according to the profitability level of crop production, four strategies were proposed (Table 11).

**Table 11.** Strategies for improving the potential of Ukrainian regions in crop production.

| Strategies | Characteristics |
|---|---|
| Strategy 1 (support strategy) | The strategy is aimed at support of regions with low productivity in the field of crop production. The biggest role should be given to the state support. As part of this strategy, diversification should be considered, and unprofitable organisations should be reoriented to other activities or growing other crops. |
| Strategy 2 (development strategy) | The strategy envisages supporting the development of regions in crop production, in particular, the provision of cheap loans, or state financing of promising projects. Enterprises should use the experience of competitors to improve product quality. |
| Strategy 3 (competition strategy) | The strategy envisages the development of measures to strengthen the competitive advantages of the region as well as enter international agro-markets. |
| Strategy 4 (leadership strategy) | The strategy envisages the development of measures to maintain the leading positions of the regions, maintaining high production results, introduction of innovative methods, and adaptation of developed countries' experience in crop production. |

Figure 7 presents the positioning of the regions of Ukraine in crop production in the proposed matrix of choice of strategy.

| Grouping of the regions by | Group III (high level) | | Volyn Chernivtsi | | | | Zhytomyr |
|---|---|---|---|---|---|---|---|
| | Group II (intermediate level) | Vinnytsia Poltava Chernihiv | Ivano-Frankivsk Lviv Rivne | Dnipropetrovsk Kyiv Odessa Kharkiv | Donetsk Zaporizhzhia | Sumy Ternopil Khmelnytsky | |
| | Group I (low level) | Kirovograd Cherkasy | Transcarpathian Luhansk | | | Mykolayiv Kherson | |
| | | VI cluster | V cluster | IV cluster | III cluster | II cluster | I cluster |
| | | Grouping of regions by the level of crop potential development | | | | | |

**Figure 7.** Positioning of the regions of Ukraine in the matrix of the choice of the strategy for improving the potential of the regions in crop production.

Four components of the agricultural potential were distinguished: (1) production; (2) innovation and investment; (3) natural resources and: (4) human resources. These components of the resource potential are inherent in the crop and stockbreeding industries. Economic and entrepreneurial potentials comprehensively characterise the set of the above components. Fuel, energy and land types of potential are classified as natural resource potential. Taking into account scientific developments, the components of the agricultural potential are systematised as follows:

(1) Production potential—a set of means of production, technical and technological production capacity, etc., used for production of agricultural products.

(2) Innovation and investment (including financial) potential—an ability to attract financial and investment resources and effectively use and distribute cash flows (current and future) to create new value through targeted integration of tangible and intangible assets of a firm to ensure its innovative development.

(3) Natural resource potential—a set of natural resources and conditions that can be used to meet the needs of the agro-industrial complex.

(4)     Personnel (labour) potential—a set of existing and potential employment opportunities for personnel who under certain conditions are able to realise their ability to work in the field of agricultural production.

Taking into account the selected components of agricultural potential, as well as the proposed strategies for improving the potential of regions in the field of crop production, a number of recommendations for building their capacity were offered (Table 12).

**Table 12.** Recommended measures of capacity-building depending on the chosen strategy for the crop.

| Components of Resource Potential | Strategy 1 | Strategy 2 | Strategy 3 | Strategy 4 |
|---|---|---|---|---|
| Production potential | 1. Ensuring a stable volume of manufactured products. 2. Detailing of the production plan. | 1. Ensuring a stable volume of manufactured products. 2. Increasing the rate of return. | 1. Consideration of export opportunities. 2. Production increase. 3. Quality control of planting raw materials and plants during their maturation. | 1. Production increase. 2. Detailing of the production plan. |
| Innovation and investment potential | 1. Involvement of specialists in the field of crop production. 2. Search for sources of funding—public and private. | 1. Cooperation with educational institutions. 2. Leasing and purchase of new processing equipment. | 1. Leasing and purchase of new processing equipment. 2. Cooperation of farms within the regions. | 1. Growing environmentally friendly innovative products. 2. Search for foreign sources of funding. |
| Natural resource potential | 1. Quality land use. 2. Use of crop by-products for fertiliser. 3. Cooperation with the peasants. | 1. Quality land use. 2. Lease of land shares from the state. | 1. Quality land use. 2. Application of soil protection technologies of tillage. 3. Land melioration and chemicalisation. | 1. Rational use of land resources. 2. Conservation of heavily eroded and sloping lands. |
| Human resource potential | 1. Professional training. 2. Increasing the number of jobs. 3. Decent wages. | 1. Improving the professional qualification level of workers. 2. Increase in wages. | 1. Improving the professional qualification level of workers. 2. Increasing the number of jobs. | 1. Training of staff abroad. 2. Increasing the number of jobs. 3. Increasing labour productivity. |

The developed matrix of the strategy selection for improving the potential of the regions of Ukraine in stockbreeding is presented in Figure 8. The X axis represents clusters of regions by the level of stockbreeding potential development, and the Y axis represents clusters of regions by the level of profitability of stockbreeding products.

Based on the comparison of the formed clusters of regions using cluster analysis and groups of regions according to the level of stockbreeding profitability, four strategies were proposed (Table 13).

Figure 9 presents the positioning of the regions of Ukraine in the matrix of the strategy selection for improving their potential in stockbreeding, and Table 14 presents the recommended capacity building measures depending on the chosen strategy in stockbreeding.

The proposed set of investment strategies for the development of agriculture in Ukraine according to the 3D-matrix is presented in Figure 10.

| Clusters of regions by the profitability level of stockbreeding | Group III (high level) | Strategy 2 | Strategy 3 | Strategy 4 | Strategy 4 |
| | Group II (intermediate level) | Strategy1 | Strategy2 | Strategy 3 | Strategy 4 |
| | Group I (low level) | Strategy1 | Strategy1 | Strategy 2 | Strategy 3 |
| | | I cluster | II cluster | III cluster | IV cluster |
| | | Clusters of regions by the level of stockbreeding potential development | | | |

**Figure 8.** Matrix of choice of strategy for improving stockbreeding potential of the regions.

**Table 13.** Strategies for improving the stockbreeding potential of the regions of Ukraine.

| Strategies | Characteristics |
|---|---|
| Strategy 1 (support strategy) | The strategy is aimed at strengthening state support for the stockbreeding sector, including the introduction of a simplified taxation system, loosening of regulation, restoring the state insurance support program, facilitating access to cheap credit resources, etc. |
| Strategy 2 (development strategy) | The strategy envisages renewal and modernisation of stockbreeding facilities, investment attraction, introduction of new forms and methods, etc. |
| Strategy 3 (competition strategy) | The strategy envisages the search for competitive advantages in the industry by improving product quality, entering the EU market, and increasing the investment attractiveness of the region. |
| Strategy 4 (leadership strategy) | The strategy envisages increasing efficiency using innovative and advanced production and processing technologies, promoting the development of diversified and innovative production structures. |

| Grouping of the regions by profitability level of stockbreeding products | Group III (high level) | Dnipropetrovsk Lviv Ternopil Kyiv | Khmelnytsky | Rivne | |
| | Group II (inter-mediate level) | Ivano-Frankivsk | Poltava Odessa Sumy | Luhansk Zaporizhzhia Donetsk Transcarpathian Kirovograd Chernivtsi | Vinnytsia |
| | Group I (low level) | Volyn | Zhytomyr Chernihiv Kharkiv | Kherson Mykolayiv | Cherkasy |
| | | I cluster | II cluster | III cluster | IV cluster |
| | | Grouping of the regions by level of stockbreeding potential development | | | |

**Figure 9.** Positioning of the regions in the matrix of choice of strategy for improving the potential of the regions in stockbreeding.

**Table 14.** Recommended measures of capacity-building depending on the chosen strategy in stockbreeding.

| Components of Potential | Strategy 1 | Strategy 2 | Strategy 3 | Strategy 4 |
|---|---|---|---|---|
| Production potential | 1. Quality control of manufactured products. 2. Increase in organic fertilisers for intra-district sales and use. 3. Concentration and specialisation. | 1. Intensification of the industry. 2. Improving the professional qualification level of workers. | 1. Rational system of herd reproduction. 2. Creating a strong fodder provision. | 1. Modernisation of the production base. 2. Concentration and specialisation. 3. Diversification of production. |
| Innovation and investment potential | 1. Improving the management of innovation processes. 2. Attracting innovative capital. 3. Participation in state innovation and investment programs. | 1. Increasing the innovative attractiveness of the region. 2. Marketing of agricultural products. 3. Conducting research. | 1. Business performance insurance. 2. Search for funding sources—public and private. 3. Modernisation of the technological provision. | 1. Implementation of mechanical processing of raw materials. 2. Implementation of socio-economic target programs. |
| Natural resource potential | 1. Increasing the number of cattle 2. Directed breeding of young cattle. | 1. Increasing the productivity of animals by expanding their diet. 2. Improvement of natural pastures. | 1. Improvement of natural pastures. 2. Increasing the average daily growth in live weight of animals. | 1. Purchase of animals from households to increase the animal amount. 2. Increasing the volume of egg harvesting. |
| Human resource potential | 1. Training of staff abroad. 2. Increasing the number of jobs. | 1. Increasing productivity. 2. Raising the level of wages. | 1. Labour cooperation. 2. Raising the level of wages. | 1. Increasing labour productivity. 2. Rational use of human resources. 3. Provision of medical services. |

A description of the proposed six strategies is presented in Table 15.

Support strategy—a strategy for regions with insufficient development, low competitiveness of agriculture, low and medium level of investment.

Conservation strategy is a strategy for regions that need to maintain and improve their positions at the national level, intensify access to new markets, and export products.

Change strategy involves adaptation to a dynamic external environment as well as to demand, and the optimisation of production processes.

Concentration strategy—a strategy of focusing on unique products, studying the world market demand, for example, for eco-products.

Growth strategy envisages an increase in production and sales, expansion of production capacity and product range. Leadership strategy envisages increasing the competitiveness of the regions through the introduction of modern management and the latest technologies.

| Groups of regions by the level of investment provision | Clusters of regions by the level of stockbreeding potential development | Clusters of regions by the level of crop potential development | | | | | |
|---|---|---|---|---|---|---|---|
| | | I cluster | II cluster | III cluster | IV cluster | V cluster | VI cluster |
| Regions with a high level of IP | I cluster | Strategy 6 | Strategy 5 | Strategy 5 | Strategy 4 | Strategy 4 | Strategy 4 |
| | II cluster | Strategy 6 | Strategy 5 | Strategy 5 | Strategy 4 | Strategy 4 | Strategy 4 |
| | III cluster | Strategy 6 | Strategy 5 | Strategy 5 | Strategy 5 | Strategy 4 | Strategy 4 |
| | IV cluster | Strategy 6 | Strategy 6 | Strategy 6 | Strategy 5 | Strategy 4 | Strategy 4 |
| Regions with an average level of IP | I cluster | Strategy 4 | Strategy 4 | Strategy 4 | Strategy 3 | Strategy 3 | Strategy 3 |
| | II cluster | Strategy 5 | Strategy 5 | Strategy 4 | Strategy 3 | Strategy 3 | Strategy 3 |
| | III cluster | Strategy 5 | Strategy 5 | Strategy 4 | Strategy 3 | Strategy 3 | Strategy 3 |
| | IV cluster | Strategy 5 | Strategy 5 | Strategy 5 | Strategy 4 | Strategy 4 | Strategy 3 |
| Regions with a low level of IP | I cluster | Strategy 3 | Strategy 3 | Strategy 3 | Strategy 2 | Strategy 1 | Strategy 1 |
| | II cluster | Strategy 3 | Strategy 3 | Strategy 2 | Strategy 2 | Strategy 2 | Strategy 1 |
| | III cluster | Strategy 3 | Strategy3 | Strategy 2 | Strategy 2 | Strategy 1 | Strategy 1 |
| | IV cluster | Strategy 3 | Strategy 3 | Strategy 2 | Strategy 1 | Strategy 2 | Strategy 2 |

**Figure 10.** Matrix for the investment strategy selection of agricultural development of Ukraine.

**Table 15.** Description of the investment strategies for agricultural development of Ukraine.

| Strategy | Type of the Strategy | Measures to Increase the Level of Agricultural Development within the Strategy Implementation |
|---|---|---|
| Strategy 1 | Support strategy | -financial recovery;<br>-constant analysis of the external environment;<br>-assessment of the bankruptcy probability;<br>-state support. |
| Strategy 2 | Conservation strategy | -refinancing;<br>-increasing income by attracting investment;<br>-providing additional insurance and guarantees. |
| Strategy 3 | Change strategy | -raising financial capital;<br>-developing measures to reduce external dependence;<br>-implementation of the latest production technologies through investments. |
| Strategy 4 | Concentration strategy | -analysis of competitors;<br>-increasing cash inflows. |

| Strategy | Type of the Strategy | Measures to Increase the Level of Agricultural Development within the Strategy Implementation |
| --- | --- | --- |
| Strategy 5 | Growth strategy | -increasing profitability of products;<br>-attraction of internal and external capital;<br>-analysis of the financial condition of the enterprise. |
| Strategy 6 | Leadership strategy | -market analysis;<br>-analysis of domestic and external demand;<br>-investing in innovative technologies. |

Source: own elaboration based on [27,75].

Analysis of scientific sources e.g., [49,50,76,77] showed that there is no consensus among scholars on the selection of agricultural development strategy. For a long time, researchers have used only some elements of the strategic approach in developing a scientific and methodological framework for managing socio-economic processes. The main ideas for choosing a strategy were laid by Ansoff [74], substantiating the fundamental differences between traditional methods of planning the development of production and commercial activities and a strategic approach to business development. The most famous in the strategic planning of agriculture is the model of Porter [78], which is based on the factors of stability of the firm's position in the market due to competitiveness and provides such competitive strategies: cost leadership, broad differentiation, optimal costs, market niche and specialization. Another well-known model of strategic planning is the matrix of the Boston Consulting Group's "growth—market share" [71], which helps to determine the strategy for the company's activities and financing to achieve leadership or profitability. As a disadvantage of this model, it should be noted that only two factors are taken into account—the relative market share and growth rates. As a limitation of the above methodologies, it should be noted that there are no criteria for the optimality of the target function for such important financial and economic indicators as product profitability, labor productivity, product indices, which are taken into account in our methodology.

Many other known models of strategic planning (McKinsey model [72], ADL/LC model [73], Hofer/Schendel model [79], Shell/DML model [80,81]) have the limitation of considering only two criteria. At the present stage, most scholars agree that when developing a strategy it is necessary to take into account the principles of an integrated and systematic approach [76,82], i.e., to take into account all important factors of external and internal environment, and to choose a strategy based on a set of indicators. The cluster analysis used in the study allowed to group the regions of the country according to the system of the most important socio-economic indicators that characterize the level of agricultural development.

Successful functioning of the agricultural complex of any country is impossible without scientifically substantiated strategic and tactical tasks and directions of their realization. In contrast to the strategies of the "blue and red" ocean, which are based on intuitive thinking, creativity, originality [83], in developing a methodology for choosing a strategy we followed the principle of scientificity, which provides analysis of the current situation and assessment of problems of agricultural industries (animal husbandry and crop production), identification of internal and external factors that affect the development and strategic goals of the activity. In addition, the proposed methodology for selecting agricultural development strategies is based on two main components: general scientific methods of cognition (methods of analysis and synthesis, methods of induction and deduction, methods of logical analysis, quantitative and qualitative analysis) and specific methods (aggregation, economic and mathematical modeling, methods of statistical and cluster analysis).

In addition, the study takes into account the regional features of the country in the field of agriculture [84], which allowed to choose strategic directions of agricultural

development of regions based on the overall strategy of economic development [85] taking into account regional specialization, different levels of development of market management system, unevenness in ensuring competitive production of agricultural industries in the regions, etc.

The proposed strategies are aimed at creating a competitive agricultural sector of the economy with an optimal and efficient structure of livestock and crop production, able to ensure broad interaction with its resource base, in order to produce high quality, profitable and competitive products and maintain the country's resource potential.

## 6. Conclusions

Many scholars and practitioners around the whole world pay considerable attention to the current state of agriculture and to the trends in it. This is due to the importance of this sector in the economies of many countries. It demonstrates the need to develop this sector. Nevertheless, the issue of the choice of the strategy for agricultural development of the regions is insufficiently explored. The reason is that in agriculture, important decisions are often taken intuitively or subjectively. This creates problems of a comprehensive assessment of the condition and opportunities for regional agricultural development. Given these facts, the current study aimed to present one of the possible solutions to this problem. The purpose of the study was to develop a scientific and methodological approach to the formation and choice of strategies to improve the agricultural potential of the country's regions.

The study substantiates selection of different strategies, taking into account regional peculiarities of agricultural development. This situation is due to differences in the values of such quantitative parameters as: resource potential of the region, profitability, investment support of the region. Using the method of cluster analysis, the regions of the country are grouped into different clusters depending on the values of the above parameters. For each of the selected groups of regions it is necessary to choose a different strategy for agricultural development. The results of the study allowed to confirm the hypothesis and solve the problem of choosing strategies to improve the resource potential of the region depending on the group to which the region belongs, depending on the level of the above parameters.

The agriculture sector in Ukraine, which has probably the highest resource potential in all of Europe, was analysed. Using cluster analysis, according to the level of development of the resource potential of crop production and the statistical data of 2018, six clusters of regions were identified. The findings of the research show that the best average level of development of the crop industry is observed in the regions that are part of the first and sixth clusters. The first cluster has the highest average values of such indicators as labour productivity at agricultural enterprises and gross harvest of cereals and legumes. The sixth cluster, which includes Vinnytsia, Kirovohrad, Poltava, Cherkasy and Chernihiv regions, is characterised by the maximum average values of crop production indices in farms of all categories and crop production per capita. The lowest development of crop production is observed in the second cluster. It has the lowest average values of three indicators: indices of crop production, crop production per capita and labour productivity. Furthermore, as a result of the cluster analysis, four clusters of regions were identified according to the level of development of the stockbreeding resource potential. The fourth cluster leads in the level of development of stockbreeding potential, because all indicators have the highest average value among the regions of Ukraine. The least-developed group is the third cluster. The regions of the second cluster are characterised by stagnation and low rates, while the indicators of the first cluster are at the average level.

In addition, according to the value of the advantage function of the profitability level of crop and livestock products in agriculture in 2018, three groups of regions of Ukraine were identified:

Group I includes regions of Ukraine with a low level of profitability of crop and livestock products. They are characterised by irrational use of resources, possible shortcomings in management, and low prospects for investment activities and the introduction of new

technologies. For these regions it is necessary to create an effective economic mechanism, based on a combination of state regulation and self-regulation, the use of new technologies, and achieving balanced price, credit and tax policies;

Group II includes regions with an average level of profitability of crop and livestock products. These regions are a good springboard for the active implementation of new tillage technologies and the use of the latest equipment and methods of raw material processing;

Group III includes the leading regions, which are characterised by high end-results of management, showing the ratio between available and used resources. The natural and economic conditions of these regions are favourable for agricultural business. The regions have significant land resources. In addition, one observes significant investment activity in Group III.

Thus, the grouping of the regions of Ukraine by the level of development of potential of crop and livestock production, as well as by the level of profitability of crop and livestock products allowed to answer the first two questions posed in the study:

A1. Based on the comparison of clusters of regions of Ukraine according to the level of development of resource potential of crop production and the level of profitability of crop production, the following strategies for improving the resource potential of regions of Ukraine in crop production are proposed: support strategy, development strategy, competition strategy, leadership strategy.

A2. Based on the comparison of clusters of regions of Ukraine by the level of development of resource potential of livestock and the level of profitability of livestock products, the following strategies are proposed to improve the resource potential of regions of Ukraine in animal husbandry: support strategy, development strategy, competition strategy, leadership strategy.

Three groups of regions were also singled out according to the level of investment provision in agriculture. The first group is characterised by a low level of investment undertakenat the expense of owners of households or enterprises. At the initial stage, they need state assistance to improve their situation for further attracting foreign investors. In turn, the second group is characterised by an average level of investment in agriculture. Attracting new investments will contribute to its development, to its rational and efficient use of funds, and the improvement of the quality of manufactured products. A high level of investment is the main feature of the third group. Farms in these areas are the basis for attracting foreign capital, expanding the range of products and their rapid entry into the global competitive agro-industrial market. Grouping of the regions of Ukraine by the level of investment support of agriculture allowed to answer the third question:

A3. Based on the comparison of clusters of regions of Ukraine according to the level of resource potential of crop and livestock and the level of investment support of agriculture, the following investment strategies for agricultural development of Ukraine are proposed: support strategy, conservation strategy, change strategy, concentration strategy, growth strategy, leadership strategy. As a result, our study confirm the main hypothesis formulated in our paper.

The choice of strategy of development of agro-branch of a region depends on three key parameters: (1) resource potential of the region; (2) product profitability; and (3) investment support of the region.

There are several contributions made by this study, both of theoretical and practical nature. First of all, this is the next voice in the discussion on development strategies in sectors of crucial importance not merely in the Ukraine, but on a global scale. Given the rather limited sources of scientific discussion on this topic, this is undoubtedly a meaningful contribution of the study and adds significant value to the theory. Secondly, four components of the agricultural potential were identified, i.e., (i) production, (ii) innovation and investment, (iii) natural resources, and iv) human resources. Taking into account the identified components, as well as the proposed strategies to improve the capacity of regions in the field of crop and livestock, a number of recommendations for capacity building depending on the chosen strategy was provided. Secondly, the main result of the

conducted study is construction of a 3D matrix of the choice of investment strategy for agricultural development based on cluster analysis of regions by the level of development of crop and stockbreeding potential and grouping of regions by the level of investments in agriculture. It allowed us to identify six investment strategies for the development of agriculture in Ukraine. This instrument may be useful when making analyses for other countries. Furthermore, the practical value of the obtained results is that they can be useful when choosing a strategy to improve the capacity of regions and making optimal management decisions at different levels. This may positively affect the performance of individual farms and agriculture in general.

Of course, the study has some limitations. The first of these is the fact that the agricultural industry in an emerging economy was analysed, where this sector is rather underdeveloped compared with the economies of developed Western countries. Another limitation of the study is the methodology used. It includes only two areas of the agricultural sector, i.e., crop production and animal husbandry, but not fisheries. Under such conditions, countries with—for example—a high level of fisheries potential will not be able to use this methodology. Given this fact, it is not clear whether the proposals which are the result of the analysis conducted may be used for analyses of more developed countries. Lastly, the analysis is based on indicators collected by the State Statistics Service of Ukraine. In other countries, where data may be collected through different methods, as well as from other sources, the method will need to be adapted. The results of the analysis are based on statistical data only for 2018, which does not make it possible to track the results in the dynamics that may affect the selection of strategy in the future. Nevertheless, despite these limitations, it is strongly believed that this study presents the real situation with regard to the potential of the agriculture resources of the regions of Ukraine and possible strategies to be implemented in the country.

The limitations described above mean that further research is needed. After all, this study is based on statistics of one year, which shows the static results, as the data in the next period will be different from the previous ones. Based on the developed methodology, it is possible to analyse the change in the position of the country's regions in the matrices of choice of strategies in the dynamics, which may be the result of further research.

**Author Contributions:** The concept and methodology were prepared by N.S. and I.K., while the detailed literature review, discussion and recommendations have been done by M.G. and N.H. The revision (review and editing) of the proposed literature review, discussion and recommendations have been done by N.S., J.V. and M.G. The funding was arranged by J.V. All authors have read and agreed to the published version of the manuscript.

**Funding:** The project was funded within the auspices of the programme of the Minister of Science and Higher Education entitled "Regional Initiative of Excellence" in 2019–2022, project number 018/RID/2018/19; amount of funding: PLN 10,788,423.16.

**Institutional Review Board Statement:** Not applicable.

**Informed Consent Statement:** Not applicable.

**Data Availability Statement:** Not applicable.

**Conflicts of Interest:** The authors declare no conflict of interest.

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
