# Peer review of "Formulation of Development Strategies for Regional Agricultural Resource Potential: The Ukrainian Case"

_resources, doi:10.3390/resources10060057_

Round 1

Reviewer 1 Report

Dear colleagues,

The article may be published after minor revisions.
The research is consistent, well documented.

For the flow of the article, I suggest you move Figure 4 (Scientific and methodological approach to the formation and selection of strategies.to improve the agricultural potential of regions) from Discussions to the chapter on methodology (3. Materials and Methods). This scheme is a flow charts, which is used in the methodology.

Also, the first part of ”6. Methodological approach to the choice of investment strategy for development of agriculture in Ukraine” is also related to the methodology chapter.

Sincerely,

Author Response

Dear Reviewer,

Thank you very much for your comments to our paper. Please be informed that we have addressed all of them and all the changes and additions (as well as movement of place in the paper) are marked yellow. The comments were:

  1. Figure 4 entitled „Scientific and methodological approach to the formation and selection of strategies to improve the agricultural potential of regions” should be moved from the Discussion section to the Materials and Methods section (point 3);

It was done.

  1. The first part of point 6 (Methodological approach…) should also be moved to the Material and Methods section.

It was done.

Please also be informed that the whole text was proofread by the native speaker.

We do believe that these improvements meet your expectations.

Best regards

Authors

Reviewer 2 Report

The paper brings research results in a choice of investment strategies for regional agriculture development in Ukrainian regions. The focus of the paper suits the scope of the journal.

The structure of the paper follows a structure of a scientific paper. However, the content of Results, Discussion, and Conclusion should be reorganized.

My comments for improvement of the paper are as follows:

  1. Literature review: add information regarding types of development strategies and research in agricultural strategies
  2. Material and methods: add information and describe input data and sources of data used in research
  3. Results: concentrate all results achieved to this section including figure 4, 5, and also section 6
  4. Remove section 6 – information move to Results
  5. Discussion: give an evaluation of the results to this section – move some parts of the conclusions here.
  6. Conclusions: some parts belong to the Discussion. Add information about future research direction.
  7. Formal remark: don´t use “we…”

Author Response

Dear Reviewer,

Thank you very much for your comments to our paper. Please be informed that we have addressed all of them and all the changes and additions (as well as movement of place in the paper) are marked yellow. The comments were:

  1. Literature review: Add information regarding types of development strategies and research in agricultural strategies.

 This information was added.

  1. Materials and methods: add information and describe input data and sources of data used in the research

It was done and this information was added.

  1. Results: concentrate all results to this section including Figures 4,5 and section 6.

  1. Remove section 6 – information move to Results

It was done.

  1. Discussion: give the evaluation of the results to this section – move some parts of the conclusions here.

This comment is connected with the next one, i.e.

  1. Conclusion: some parts belong to the Discussion. Add information about future directions of the research.

It was done. We added information about the future directions of the research.

  1. Formal remark: don’t use “we”.

It was done.

Please also be informed that the whole text was proofread by the native speaker.

We do believe that these improvements meet your expectations.

Best regards

Authors

Reviewer 3 Report

The title is suitable for the study.

In the summary, originality remains to be put at the end.

The introduction is very well structured. I suggest that they reinforce the GAP with more literature.

The literature review needs to be developed.

I recommend that they formulate the hypotheses in the literature review.

I also recommend that at the end of the literature review, place the research model with the hypotheses.

In the results, they have to indicate whether the hypotheses are confirmed or not.

The methodology needs to be more detailed. Should they justify the importance of using this method in this study? Support with literature. Authors who applied the same method?

Data collection needs to be better explained.

The results seem robust to me.

In discussing results, authors have to compare their results with the literature.

The conclusion needs to be developed. I suggest the following structure:

  1. Remember the objective of the study
  2. Main findings
  3. Theoretical implications (need to improve)
  4. Practical implications (needs to be refined)
  5. Study limitations (need to improve)
  6. Future lines of research (not available)

The references are appropriate to the study and current.

Good luck with the publication!

Author Response

Dear Reviewer,

Thank you very much for your comments to our paper. Please be informed that we have addressed all of them and all the changes and additions (as well as movement of place in the paper) are marked yellow. The comments were:

  1. Introduction is very well structured. I suggest to reinforce the gap with more literature.

It was done.

  1. The Literature review needs to be developed.

It was done.

  1. I recommend to formulate the hypotheses at the end of literature review as well as research model

Thank you for this comment. We have analyzed deeply how to address it, but finally, we came to the conclusion that we will leave it as it is now. Please be informed that none of the other reviewers (there were 3 reviewers totally) requested to add hypotheses. In addition, due to the nature of the paper which is to some extent a conceptual study we believe it is not necessary (because of this, we didn’t add a research model in this part too as well as information on hypothesis in the Results). We do hope you accept our arguments.

  1. The methodology needs to be more detailed. Importance of using this method in the study? Support with literature? Authors who applied the same method.

It was done.

  1. Data collection needs to be better explained.

It was done.

  1. In discussing details, authors have to compare their results with the literature.

To some extent that’s a conceptual study which proposes some possible development strategies to be applied in the agricultural sector based on analysis of two dimensions: crop production and animal husbandry. It would be rather hard to compare the results with findings of other researches.

  1. The conclusion needs to be developed. I suggest the following structure

Objective of the study

Main findings

Theoretical implications (need to be improved)

Practical implications (need to be refined)

Study limitations (need to be improved)

Future lines of research (not available)

It was done.

We do believe that these improvements meet your expectations.

Please also be informed that the whole text was proofread by the native speaker.

Best regards

Authors

Round 2

Reviewer 3 Report

The study was improved, although some suggestions were not incorporated in the work.

- I recommend to formulate the hypotheses at the end of literature review as well as research model
- In the results they have to indicate whether the hypotheses were confirmed or not.
- In discussing details, authors have to compare their results with the literature.
- The theoretical implications are practically nonexistent.
- Study limitations and future lines that are not very well developed.

Author Response

Dear reviewer,

All comments have been revised and highlighted in green.
